# Wind lidars reveal turbulence transport mechanism in the wake of a tree

Nikolas Angelou[1], Jakob Mann[1], and Ebba Dellwik[1]

[1]Wind Energy Department, Technical University of Denmark (DTU), Roskilde, 4000, Denmark

**Correspondence:** nang@dtu.dk

**Abstract.** Solitary trees are natural land surface elements, found in almost all climates, yet their influence on the surrounding air flow is poorly known. Here we use state-of-the-art, laser-based, remote sensing instruments to study the turbulent wind field in the near-wake region of a mature, open-grown oak tree. Our measurements provide for the first time a full picture of the mixing layer of high turbulence that surrounds the mean wind speed deficit. In this layer, we investigate the validity of the eddy-viscosity hypothesis, a hypothesis commonly used in modelling the turbulence transport of momentum and scalars in the atmosphere. We find that the momentum fluxes of the streamwise wind component can be adequately predicted by the transverse gradient of the mean flow. Using the mixing-length hypothesis we find that for this tree the corresponding turbulence length scale in the mixing layer can be approximated by one, height-independent value. Further, the laser-based scanning technology used here was able to accurately reveal three-dimensional turbulent and spatially varying atmospheric flows over a large plane, without seeding or intruding the atmospheric flow. This capability points to a new and more exact way of exploring the complex earth-atmosphere interactions.

## 1 Introduction

Solitary trees are common elements on Earth's surface, planted or naturally grown, in urban and rural landscapes, as well as in tundras and grasslands. Since trees are exceptionally efficient at extracting momentum from the wind (Lee et al., 2014; Dellwik et al., 2019), they can cause a significant effect in the near-surface atmosphere. This effect of the wind-tree interaction explains why trees are used as engineering elements in shelter-belts to reduce the mean wind (Miller et al., 1974), decrease traffic noise (Kragh, 1981), improve crop productivity (Campi et al., 2009) and mitigate surface erosion (Miri et al., 2017). Because trees strongly reduce the wind speed, they also cause a downstream increase of the wind gradients. This, in turn, contributes to generation of turbulence, and thereby trees also have a strong effect on the downstream turbulent transport of scalars (temperature, humidity, gases and particles) and momentum. An accurate description of the wind-tree interaction is also important for our understanding of the biological effect that wind has on trees, which determines their growth (Telewski, 1995) and influences their physiological response (de Langre, 2008) and their ability to reduce particulate air pollution (Chen et al., 2017). Yet, despite its fundamental importance, the scientific description of trees' interaction with the atmosphere is surrounded by large uncertainties.

Trees consist of multi-scale flexible elements, branches and leaves, that respond dynamically to the wind (Gosselin, 2019). Due to these characteristics, it is difficult to evaluate whether simplified representations of trees both scaled in wind-tunnels (Bai et al., 2012) and in numerical experiments (Gross, 1987; Gromke and Ruck, 2008) realistically reproduce the effects of natural trees. However, also atmospheric outdoor experiments into how trees influence the wind field have been limited by our inability to accurately observe the spatial variability in the wind field. By use of conventional *in situ* wind observations, the
retrieved data represents the flow properties at one point, which is insufficient for the complete characterization of a complex flow where a high-gradient wind field can occur.

Over the last decade a new possibility to study atmospheric flows has emerged based on the synchronous operation of three scanning Doppler wind light detection and ranging (lidar) instruments to simultaneously probe an air volume (Mikkelsen et al., 2017). Their design enables the measurement of the three-dimensional wind vector while rapidly scanning air volumes
up to a distance of approximately 150 m (Sjöholm et al., 2018). This remote sensing technique does not require release of tracer particles, which is necessary when characterizing complex flows in wind tunnels. Instead, the technique is based on the detection of the Doppler shift of the backscattered laser light by naturally occurring aerosols (Henderson et al., 2005). This measurement methodology is especially useful in monitoring complex flow patterns, as in the case of wind over forests or hills (Mann et al., 2017), or in the case of wakes from wind turbines or surface obstacles, such as trees, since the whole flow field
can be scanned. Here, we demonstrate a new application of this measurement technique, where the turbulent transport in the wake of a single tree is made visible and quantified. By this extension, it is possible to retrieve spatially distributed statistics of the wind in the real atmosphere that were previously only attainable in idealized numerical simulations or scaled wind tunnel studies. This direct measurement of the second order moments enables the validation of the contested (Schmitt, 2007) and much-debated (Finnigan et al., 2015) eddy-viscosity hypothesis, which is a corner-stone in the simulation of weather (Powers
et al., 2017), dispersion of pollution (Jeanjean et al., 2015), siting of wind turbines (Landberg et al., 2003) and atmosphere-canopy interaction (Cowan, 1968; Bache, 1986; Sogachev et al., 2012).

## 2 Methodology

### 2.1 Experimental Setup

The tree in focus in this study is an open-grown oak tree (*Quercus robur*), located 60 m from the shoreline of the Roskilde
fjord (Denmark) (32U, 694598E, 6175776N) and 2.6 m above the sea level (Fig. 1a-b). Its shape is common for solitary trees, and it has a height $H = 6.5$ m and width of 8.5 m, which are within the typical range for open-grown oak trees (Hasenauer, 1997). The height of the tree is used to normalize all the spatial dimensions, which are denoted by the ˆ symbol. The study of the wind field at the lee side of the tree is performed using measurements from a multi-lidar system denoted as *short-range WindScanner*. A short-range WindScanner system consists of three separate scanning wind lidar instruments, denoted
in this document as $WS_1$, $WS_2$ and $WS_3$, developed in the Wind Energy department of the Technical University of Denmark (Mikkelsen et al., 2017). Each instrument is a mono-static, coherent, all-fiber, infrared, continuous-wave (cw) wind lidar. The elevation and azimuth angles of the laser output is steered by an optical scanner head (Sjöholm et al., 2018). The scanner head

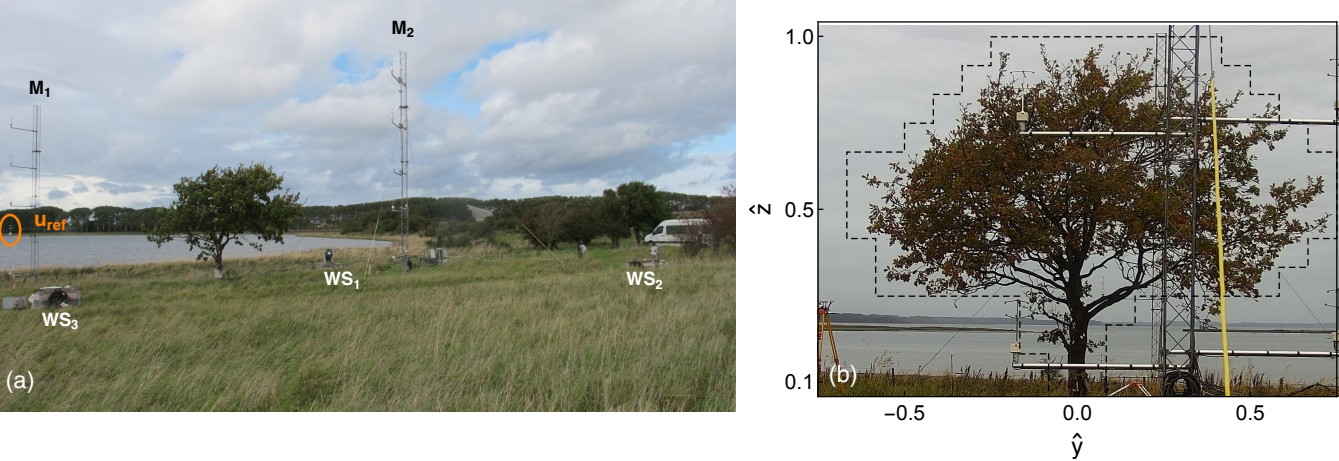

**Figure 1. Photographs of experimental setup**. **(a)** Area surrounding the oak tree, where the three scanning wind lidars ($WS_1$, $WS_2$ and $WS_3$) and the two meteorological masts ($M_1$ and $M_2$) were located. **(b)** A view from the lee side of the frontal area of the tree. The $y$ and $z$ axes are normalized by the tree height.

consists of two individually-rotating prisms that can direct the line-of-sight anywhere within a cone with a 120° base-angle. The backscatter light from the atmosphere for each line-of-sight is collected by a 3" telescope and detected using a homodyne configuration (Abari et al., 2014). For the needs of this study, we define a right-handed coordinate system whose origin is at the base of the tree's stem and the $x$-axis is aligned to the predominant wind direction during the period of the experiment (positive $x$-axis pointing towards 110° relative to the geographic North).

### 2.1.1  Scanning mode

The position of the three lidars, displayed in Fig. 2a, was selected based on the criteria that i. the instruments should be as close as possible to a measuring position in order to ensure short probe lengths, ii. the measuring area should be within the field-of-view of each lidar, and iii. the direction of the three line-of-sights should at every scanning location enable the estimation of the three-dimensional wind vector. The latter is not fulfilled closest to the ground where the line-of-sight of the laser beams are close to being horizontal, making it impossible to resolve the vertical component of the wind vector.

The lidars were programmed to acquire measurements within a vertical plane, that was normal to the $x$-axis at a distance of 8.5 m ($\hat{z}$=1.3) from the tree towards the leeward direction. At that distance, the terrain is elevated by 0.25 m in comparison to the location of the tree. Due to the small difference in elevation and the relatively short distance between the tree and scanning plane, the geometry of the tree and the wind measurement locations are presented relative to the corresponding local ground level. The plane was synchronously scanned using a trajectory that consisted of 30 vertical lines that extended from 1.5 m ($\hat{z}$=0.2) to 16 m ($\hat{z}$=2.5). The lines spanned from −7.25 m to 7.25 m (−1.1 < $\hat{y}$ < 1.1) across the $y$-axis forming a rectangle, whose height and width were equal to $2.5H$ and $2.2H$, respectively. Due to the close distance from the tree and the dimensions of the plane the wake was always within the scanning area for the wind direction sector used in this study. The three laser

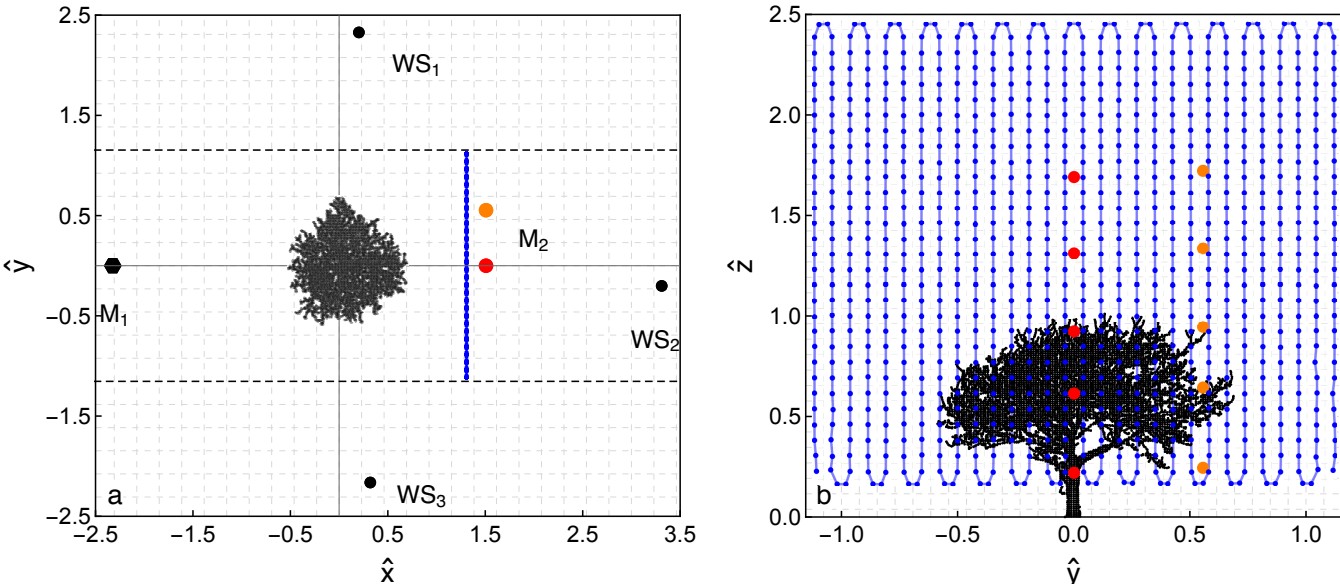

**Figure 2. Experimental setup**. **(a – b)**, Drawings of the top (left) and the front (right) views of the experimental setup, where the locations of the tree, the short-range WindScanners ( $WS_1$, $WS_2$ and $WS_3$), the scanning pattern (blue) and the locations of the sonic anemometers on the up- ($M_1$) and down-wind ($M_2$) meteorological masts, are depicted. The locations of the 10 sonic anemometers on the opposing booms on the $M_2$ mast are indicated with red and orange color, respectively.

beams were following the path of the vertical lines with an alternating direction of motion, starting from the lowest height (Fig. 2b). The scanning duration of each line was 710 ms, which, including the transition time between two neighbouring lines (90 ms), resulted in a scanning duration of the whole rectangle of 24 seconds. Following the completion of one scanning plane,

80 the three laser beams were returning to the original location of the first measurement within 2 seconds. Based on these features, approximately 25 iterations of the scanning pattern were performed per 10-minute period.

### 2.1.2 Grid

The data acquired within an iteration of the scanning trajectory was grouped in square grid cells with a dimension of 0.5×0.5 m. The scanning speed (20 ms$^{-1}$) and the sampling rate (205 Hz) of the instruments resulted in acquiring on average five Doppler

85 spectra per grid cell per scanning pattern iteration. The Doppler spectra acquired in each grid cell were averaged in order to decrease the variance of the noise floor. Finally, the estimation of the Doppler frequency was performed using a method based on the median of the accumulated energy in the spectrum. This method has been proven to be less biased by noise (Angelou et al., 2012a) and to produce accurate first and second order statistics (Held and Mann, 2018).

In the cw wind lidars, as the ones used in this study, the probe length is dependent on the optical properties of the telescope

90 (i.e. effective radius of the lens) and the focusing distance. The dimension of the probe length is based on the distribution of the intensity of the focused light along the line-of-sight, which is approximated by a Lorentzian function (Sonnenschein and

Horrigan, 1971). Based on the configuration of the experimental setup, the probe length of each lidar along the line-of-sight varied between 0.25 and 1.87 m, depending on the measuring position. This range in combination with the tilt and azimuth angles of each line-of-sight of the scanning wind lidar results to a theoretical spatial resolution of 0.55 – 1.45 m, 0.49 – 0.89 m and 0.02 - 1.08, for the $x$-, $y$- and $z$-axis, respectively (Angelou, 2020). In the area close to center of the scanning plane $-0.7 < \hat{y} < 0.7$, where the wake is expected to be found the spatial resolution varies between 0.55 – 0.96 m, 0.49 – 0.58 m and 0.02 - 0.34 m, for the $x$-, $y$- and $z$-axis, respectively. By assuming that the spatial variation of the wind speed along the streamwise direction is negligible in those scales, the spatial resolution is comparable to the grid cell size 0.5 m $\times$ 0.5 m used in this study.

### 2.1.3  Sonic anemometers

In addition to the lidar measurements, *in situ* observations from multiple sonic anemometers were used as a reference of the up- and down-wind conditions in this study. Two meteorological masts used, were equipped with sonic anemometers (uSonic-983 Basic, Metek Gmbh, Hamburg, DE), installed at five different heights 1.5 m ($0.23H$), 4.0 m ($0.62H$), 6.0 m ($0.92H$), 8.5 m ($1.31H$) and 11.0 m ($1.69H$), above ground level (a.g.l.). The two masts denoted as $M_1$ and $M_2$ in Fig. 2a-b were located in two anti-diametric locations from the tree. The positions were chosen in order to allow the simultaneous monitoring of the wind conditions in both the windward ($M_1$) and the leeward ($M_2$) directions, when the wind direction was 290°. The sonic anemometers were installed on booms pointing towards the direction of 200° relative to North on both masts. To get a higher coverage of the complex wind field in the tree wake, the $M_2$ mast was instrumented with five additional sonic anemometers, on opposing booms pointing towards 20° relative to North. Based on the length of the booms, an array of ten sonic anemometers was formed in the $M_2$ mast, with a width of 3.6 m ($0.55H$) and a height of 10 m ($1.53H$). A flow distortion correction algorithm was applied to all the sonic anemometer high-frequency data (20 Hz) following the method described in Peña et al. (2019).

## 3  Data analysis

### 3.1  Data Selection and Post Processing

The field test using the scanning wind lidars was performed over a period of one month (October 2017), during which the wind direction was not suitable for our analysis the majority of the time. Here, we focus on a 3-hour period (13.30 - 16:30, 25-10-2017), during which the wind originated from the direction 270°-310°. In this sector, the streamwise wind component $u$ is approximately aligned to the $x$-axis of the coordinate system and the wake of the tree is expected to be within the scanning plane. The ambient mean wind speed varied between 7 – 15 ms$^{-1}$, in all heights, except at the location of the lower sonic anemometer (1.5 m) which reported relatively lower wind speeds (Fig.A1 in Appendix A). In order to assess the stationarity of the free flow, we define the time scale $\tau_H = u_\star / H$, where $u_\star = (\langle u'w' \rangle^2 + \langle v'w' \rangle^2)^{1/4}$ is calculated from the 4 m sonic anemometer at the $M_1$ mast and $H$ is the tree height. Using the mean of the friction velocity (0.4 ms$^{-1}$), $\tau_H = 16.25$ s. Over this time scale, a linear fit of the wind speed versus time showed a slope of less than 0.001 ms$^{-1}$ per $\tau_H$, and the time series of the free flow is, therefore, considered to be stationary.

The local atmospheric conditions were characterized by neutral stability, as the height normalized by the Obukhov length $L$ values, the so-called stability parameter, were found within the range $0 < z/L < 0.01$. The Obukhov length $L$ was derived using the 10-minute statistics based on the measurements of the windward sonic anemometer at 4 m, through the expression:

$$L = -\frac{T_o}{\kappa g} \frac{u_\star^3}{Q_o},$$

(1)

where $T_o$ is the temperature at a height $z$, $\kappa$ is the von Karman constant, $u_\star$ is the friction velocity defined by $u_\star = (\langle u'w' \rangle^2 + \langle v'w' \rangle^2)^{1/4}$, $Q_o = \langle \theta'w' \rangle$ is the surface virtual temperature flux, here we use the sound virtual temperature measured by the sonic anemometer, and $g$ is the gravitational acceleration (Wyngaard, 2010).

The sonic anemometer measurements at 4 m, on the $M_1$ meteorological mast, which approximately corresponds to the height of the center of the crown, were used as reference of the horizontal wind speed and direction (Fig. 1a). Subsequently, the lidar data of each iteration of the scanning pattern, was grouped according to their corresponding 26-second mean wind direction of the period, when it was acquired. First and second order moments of the wind vector components in the lee side of the tree were estimated, using the ensemble of the normalized data acquired by the individual scanning pattern iterations, when the upwind direction was between 282° - 287° (Fig.A2a in Appendix A). This wind direction sector was selected since it was the one with the most frequent occurrences (101 scanning pattern iterations). The wind speed during these occasions varied between 6.5 - 10.5 ms$^{-1}$, with a mean of 8.05±0.80 ms$^{-1}$ (Fig.A2b in Appendix A). The calculation of the wind statistics, and especially 2$^{nd}$ order moments, may still yield theoretically satisfactory results, with relatively low systematic and random errors, even using few measurements, given that certain criteria are fulfilled. According to the work of Lenschow et al. (1994) these criteria are based on the high sampling frequency and the relation between the disjunct sampling time (here defined as the time between the scanning pattern iterations) and the time integral scale of the fluctuating quantity. We elaborate more on this issue in Section 5. Prior the estimation of the wind statistics, two post-processing steps were performed. First, a filter was applied on the wind lidar data. The filtering was based on the calculation of the inner and outer fence of the distribution of $w$ in each grid cell. Those wind vector estimations with a vertical component outside the two fences in each grid cell were treated as outliers and were not included in the analysis. Afterwards the wind speed measurements of each iteration, were normalized by the corresponding 26-second mean wind speed ($u_{ref}$), acquired at 4 m at the upwind mast. The normalized quantities are indicated with the ˆ symbol and presented using the Reynolds decomposition, where the mean and turbulent fluctuations are denoted by the $\langle \rangle$ and $'$ symbols, respectively.

## 4 Results

### 4.1 Vertical profiles

In Fig. 3a–f, we show the vertical mean and variance of the longitudinal wind speed and the turbulent momentum transport (flux) profiles measured by the wind lidars (blue lines) and the sonic anemometers (red and orange filled circles for $\hat{y} = 0$ and $\hat{y} = 0.5$, respectively) located approximately 1.5 m downwind of the scanning plane (Fig. 3a and e). It is observed that the vertical profiles of the mean streamwise component measured by the lidars are consistent with the sonic anemometers,

regardless the vertical and spanwise distance. The difference between the individual first order moment estimations of the two instruments is varying between -2.0 % and 8.0 %, depending on the location. The smallest differences, lower than 2%, are found in the lowest (i.e. $z = 1.5$ m) and highest (i.e. $z \geq 8.5$ m) heights. In the wake the differences are increased to 3.5 – 8.3%. Overall, a mean absolute difference of 3% is found between the two measuring instruments. As far as it concerns the second order moments, the vertical trends of variances and covariances of measured by the sonic are also captured by the wind lidars (see Figure 3 b–d and g–i). The relative error in the longitudinal variance varies between -6% and -58%. The maximum relative error is found in the sonic anemometer at the center of the wake (sonic anemometer at 4 m, south boom at $M_2$), where the turbulence is very low. The observed underestimation of $\langle \hat{u}' \hat{u}' \rangle$ is attributed to the probe length of wind lidar, which operates as a low-pass filter which attenuates the high frequency fluctuations of the wind (Angelou et al., 2012b). As far as it concerns the momentum fluxes, we find overall relative error values that are up to 128% above the tree, but the error in the wake is limited to 20%. This does not include the vertical momentum fluxes at the lowest height (1.5 m), which, due to the geometry of the experimental setup, are found to be very sensitive to random noise. Furthermore, we have not included the vertical momentum flux measurement at 4 m at the north boom of the $M_2$ mast, since the correlation between the longitudinal and vertical component was found to be equal to 0 $m^2 s^{-2}$, leading to an unspecified relative error. The uncertainty on the first and second order statistics is estimated by splitting the ensemble in four equally sized subsets and subsequently calculate the ratio of the standard deviation of each moment with the square root of $n - 1$ ($n = 4$). The values are presented in Fig. 3 with a blue shaded area and error bars for the case of the scanning wind lidar and sonic anemometer measurements, respectively. An increasing measurement uncertainty with decreasing distance to the ground is observed for the vertical momentum in the height range of $0 < \hat{z} < 0.5$ (Fig.3d and i), which can be explained by the low elevation angle of the lidars' line-of-sight. A low elevation angle makes the measurement of the horizontal velocity component relatively more accurate, whereas it deteriorates the accuracy of the vertical velocity component.

## 4.2 Strong fluxes collocated with sharp gradients

The lidar measurements reveal the statistics of the three-dimensional wind vector in all the measurement points of the scanning plane (Fig.2b). In this near-wake region of the tree, the shape of the deficit closely resembles the shape of the crown (see dashed line in Fig. 4a). The asymmetry of the deficit in the wake can be explained by heterogeneities in the tree's plant area density. At the height of the reference upwind speed ($\hat{z} = 4.0/6.5 = 0.6$), the minimum normalized wind speed observed in the wake is measured to be 27% of the upwind speed. This is more than what was observed for the summer period, where the minimum was closer to 10% Dellwik et al. (2019), indicating that the tree at the time of the experiment in late October was in abscission phase and that a significant amount of leaves had been lost.

The edges of the wake are characterized by an increase of the streamwise variance $\langle \hat{u}' \hat{u}' \rangle$. This increase reveals a thin interface layer of high turbulence, with a width between $1 - 2$ m ($0.15 - 0.3$ $H$), extending along the periphery of the tree's crown. This layer corresponds to the area where momentum transfer between the free wind and the center of the wake takes place and it is a typical feature of wakes generated by porous bodies (Huang et al., 1995; Bai et al., 2012). This area is characterized by strong wind speed gradients (Fig. 4b-c) that are co-located with strong turbulence that transports momentum towards the wake

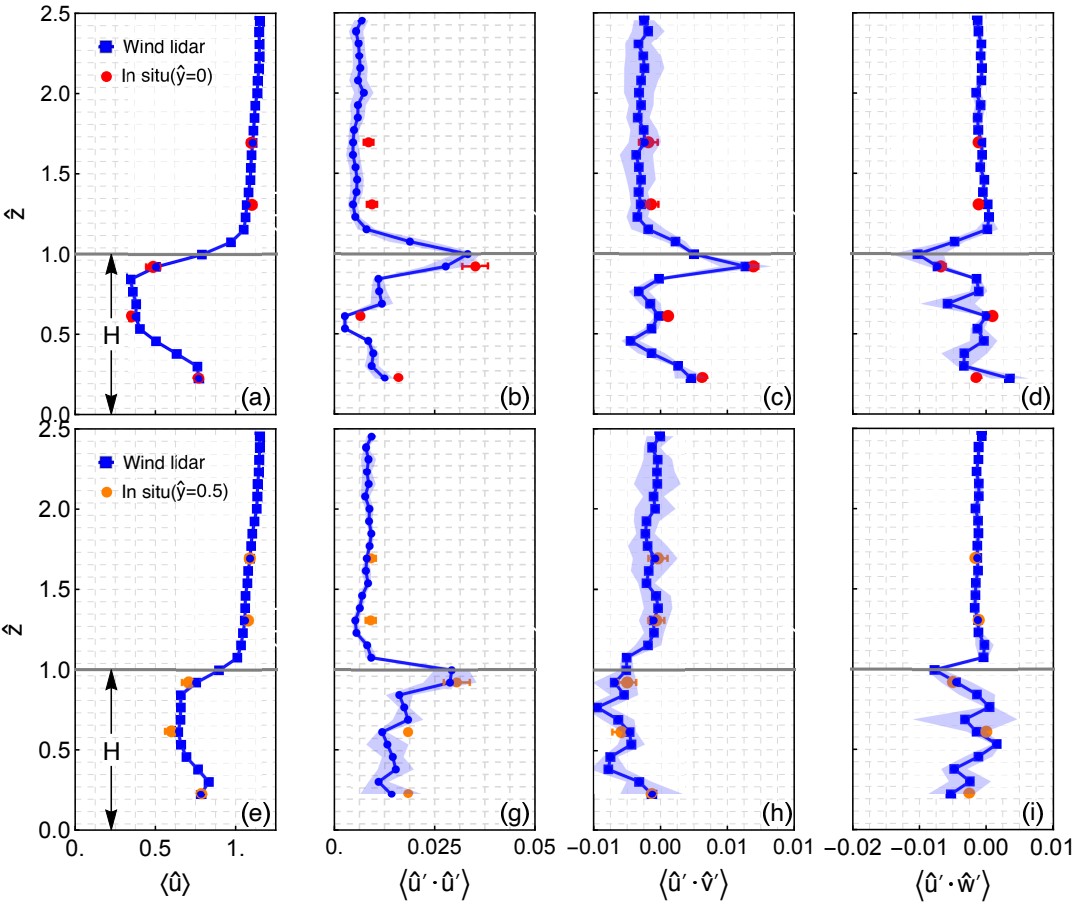

**Figure 3.** The vertical profile of the normalized mean streamwise wind speed $\langle \hat{u} \rangle$ (a, e), the longitudinal variance $\langle \hat{u}'\hat{u}' \rangle$ (b, g), the horizontal $\langle \hat{u}'\hat{v}' \rangle$ (c, h) and the vertical $\langle \hat{u}'\hat{w}' \rangle$ (d, i) momentum fluxes, measured by the short-range WindScanner (blue) and downwind sonic anemometers in the center of the wake $\hat{y} = 0$ (top plots) and at $\hat{y} = 0.2$ (bottom plots). The blue area corresponds to the estimated values of the statistical uncertainty of each quantity (see more on Methods). The horizontal dashed line in (b) and (d – f) corresponds to the tree height.

center (Fig. 4e-f). The gradients were estimated by first calculating the forward difference of the mean longitudinal wind speed between neighbouring grid cells in the $y$ and $z$ direction, and subsequently by finding the corresponding value in the coordinates of each cell using linear interpolation. The horizontal transport of streamwise momentum $\langle u'v' \rangle$ (Fig. 4e) has alternating signs on the left and right sides of the tree indicating the inwards transport and it is approximately half of the vertical transport of streamwise momentum $\langle u'w' \rangle$ measured above the tree (Fig. 4f).

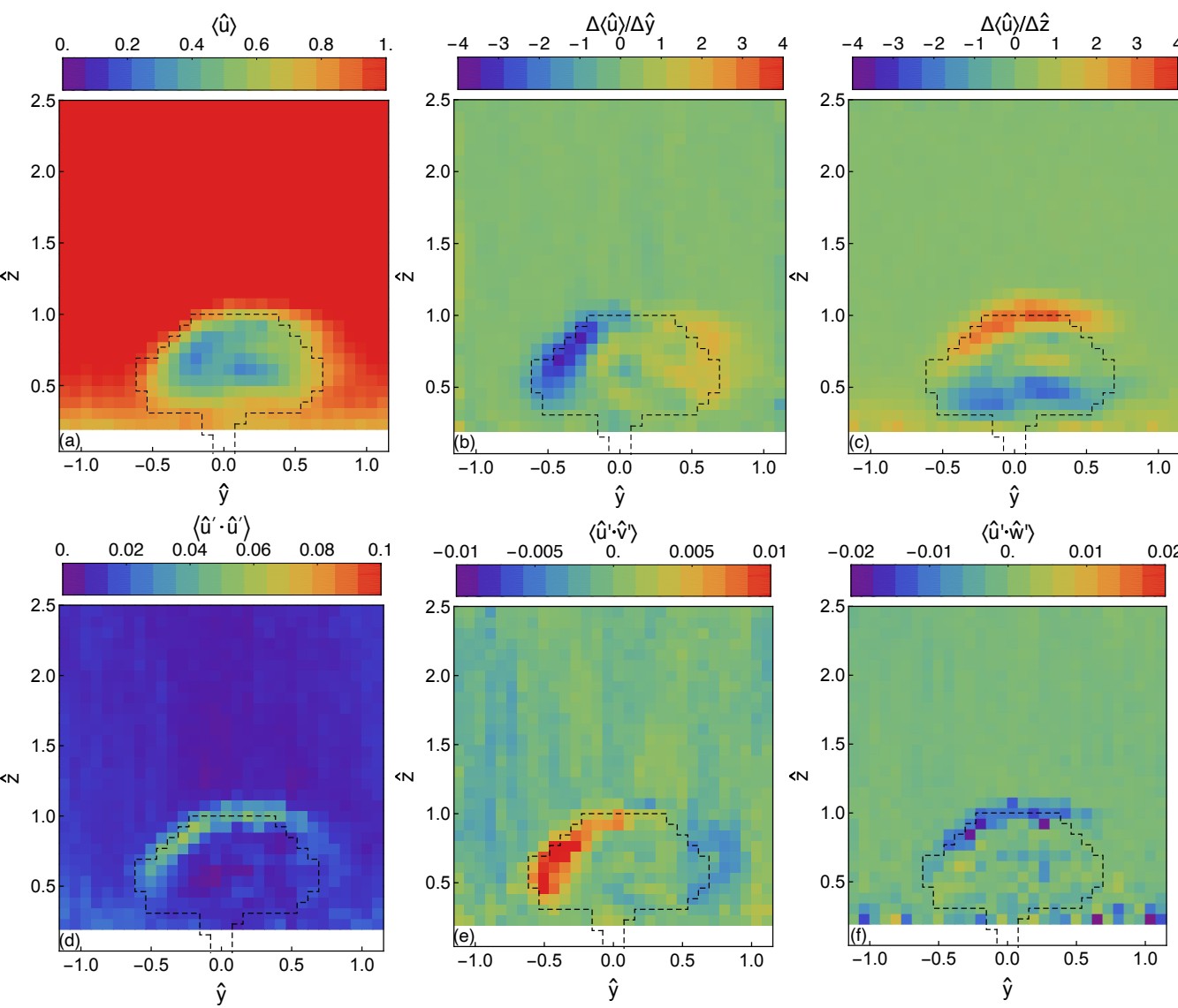

**Figure 4. Normalized first and second moments of the wind in the lee side of the tree.** (a), mean streamwise component $\langle \hat{u} \rangle$, (b), spanwise $\Delta \langle \hat{u} \rangle / \Delta \hat{y}$ and (c), vertical $\Delta \langle \hat{u} \rangle / \Delta \hat{z}$ gradient of the mean streamwise component, (d), streamwise variance $\langle \hat{u}' \hat{u}' \rangle$, (e), horizontal $\langle \hat{u}' \hat{v}' \rangle$ and (f), vertical $\langle \hat{u}' \hat{w}' \rangle$ covariances of the longitudinal wind component. The dashed line in all figures represents the periphery of the tree.

## 4.3 Test of the eddy-viscosity hypothesis

The turbulent atmosphere contains motions at a large range of scales, and all predictions of its behaviour rely on simplifying models on how the smaller scales affect the momentum and scalar transport (Launder and Spalding, 1972). The most commonly used simplification for small-scale atmospheric transport is based on the eddy-viscosity hypothesis, which expresses the relation between the momentum fluxes and the local mean gradient (Boussinesq, 1877; Pope, 2000). According the eddy-viscosity hypothesis the deviatoric part of the Reynolds stress tensor is related to the mean strain as:

$$\langle u_i' u_j' \rangle - \frac{1}{3} \langle u_k' u_k' \rangle \delta_{ij} = -\nu_T \left( \frac{\partial \langle u_i \rangle}{\partial x_j} + \frac{\partial \langle u_j \rangle}{\partial x_i} \right),$$
(2)

where $\nu_T$ is the eddy diffusivity which is a property of the flow and the term $\left( \frac{\partial \langle u_i \rangle}{\partial x_j} + \frac{\partial \langle u_j \rangle}{\partial x_i} \right)$ represents the mean strain.

Although it is widely used in atmospheric modelling (*e.g.* Kaimal and Finnigan, 1994), the general applicability of the eddy-viscosity hypothesis is contested on theoretical grounds (Pope, 2000). Further, Schmitt (2007) claimed that it is almost never valid based on results from highly advanced numerical simulations around obstacles and in shear flows. A different argument

against the validity of the eddy-viscosity hypothesis concerns that it is insufficient for predicting the atmospheric transport of scalars and momentum, since the local gradient and eddy viscosity only determine a minor part of the total turbulent transport. This argument has been used to explain observed counter-gradient fluxes inside dense forest canopies (Denmead and Bradley, 1985), by underlining that the main transport mechanism for scalars and momentum instead is the large-scale eddies (Raupach et al., 1996; Finnigan, 2000; Brunet, 2020). In this case, the atmospheric transport would not be successfully captured by

an eddy-viscosity parameterization. In sparse canopies, characterized by three-dimensional complexity, the validity of the eddy viscosity hypothesis is also disputed (Finnigan et al., 2015). In this study we focus on the transport mechanism of the longitudinal momentum. For this purpose we construct a momentum vector from the following two components:

$$\langle u_1' u_i' \rangle = \nu_T \frac{\partial \langle u_1 \rangle}{\partial x_i}, \text{ where } i = 2, 3$$
(3)

With the above equation we want to express the relation between the momentum flux and mean gradient. In this equation,

the along wind gradients of the vertical $\frac{\partial \langle u_2 \rangle}{\partial x}$ and transverse components $\frac{\partial \langle u_3 \rangle}{\partial x}$ are considered to be negligible. We base this assumption on the estimated along wind gradients ($\frac{\partial \langle u_2 \rangle}{\partial x}$ and $\frac{\partial \langle u_3 \rangle}{\partial x}$) between the wind lidar and sonic anemometer measurements at the 10 locations where sonic anemometers were found on the $M_2$ mast. The values that we find are one order of magnitude lower than the values of the transverse gradient and therefore, a significant bias should not be expected by disregarding the along wind gradient.

In Fig. 4 b–c, we can visually observe that the areas with high gradients are characterized by also strong momentum fluxes with an opposite sign, as Eq. 3 requires. We investigate this further by selecting grid points with high $\langle u'u' \rangle$ comparing to the undisturbed flow (for more information regarding the grid selection we refer to the Appendix B). We chose these grid cells both because they represent the area where the mixing takes places. In these grid cells, the angle of the gradient vector (Fig.5a) is radially pointing outwards from the center of crown, at the opposite direction of the vector of the momentum transport (Fig.5b).

In 90% of the selected grid cells the difference between the two vector directions is less than 30°, with a mean difference equal

to 177° and a standard deviation of 22° (Fig.5c). Using the same criterion as Schmitt (2007), we find that the observed relative direction of the two vectors supports the validity of the eddy-viscosity hypothesis.

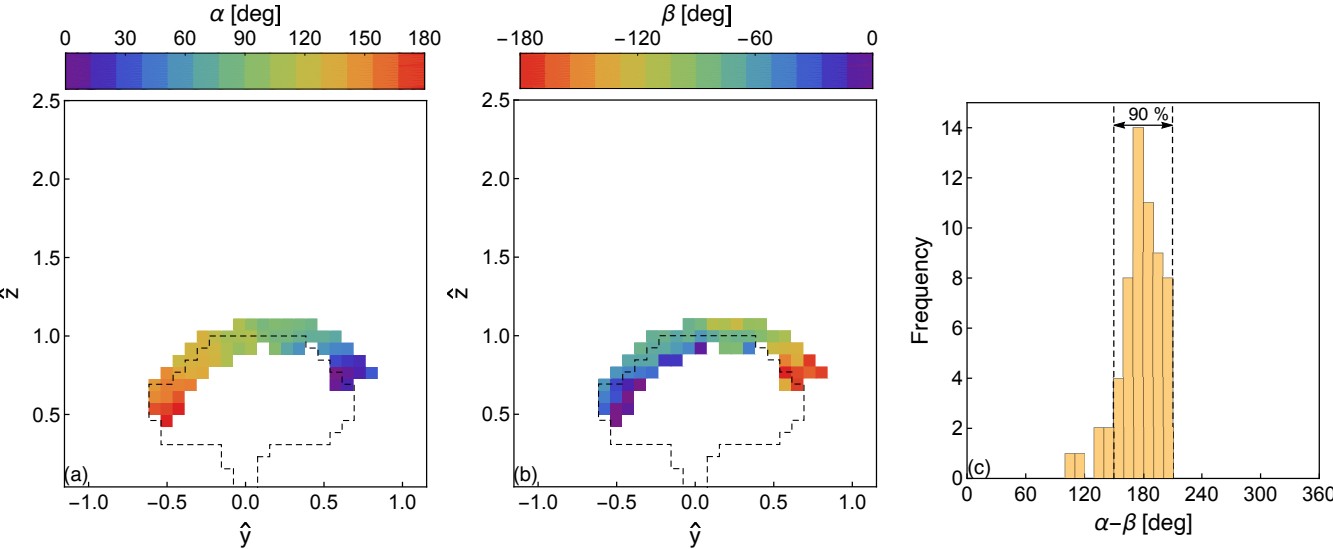

**Figure 5. Direction of the mean gradient and momentum flux of the transverse wind speed vector.** Direction of (a) the mean gradient $\left|\left(\frac{\partial\langle u\rangle}{\partial y}, \frac{\partial\langle u\rangle}{\partial z}\right)\right|$ and (b) the covariance $\langle u'_1 u'_i\rangle$ vectors. The dashed line in (a) – (b) represents the periphery of the tree. (c), Histogram of the direction difference between the two vectors. The dashed lines highlight the range where the two vectors have a difference less or equal to $\pi/6$, which in this study is used as a validity index of Boussinesq's eddy-viscosity hypothesis.

## 4.4 A small and near constant turbulence mixing length

The simplest parameterization of the eddy viscosity $\nu_T$, is constructed based on Prandtl's *mixing-length* hypothesis (Prandtl,
1925), which states that the eddy-viscosity in an air volume is equal to the absolute local gradient of $u$ times the square of a turbulence mixing length scale $l_m$

$$\nu_T = l_m^2 \left|\left(\frac{\partial\langle u\rangle}{\partial y}, \frac{\partial\langle u\rangle}{\partial z}\right)\right|. \tag{4}$$

This length scale describes the characteristic distance over which an air parcel keeps its original properties and it is an analogy to the mean free path in the statistical mechanics understanding of molecular viscosity. The concept can also be visualized as
a characteristic size of the dominant turbulent eddy responsible for the mixing of the flow. By substituting equation (4) into equation (3), the mixing length can be expressed as a function of the length of the covariance and gradient vectors

$$l_m = \frac{\sqrt{|\langle u'_1 u'_i\rangle|}}{\left|\left(\frac{\partial\langle u\rangle}{\partial y}, \frac{\partial\langle u\rangle}{\partial z}\right)\right|}, \tag{5}$$

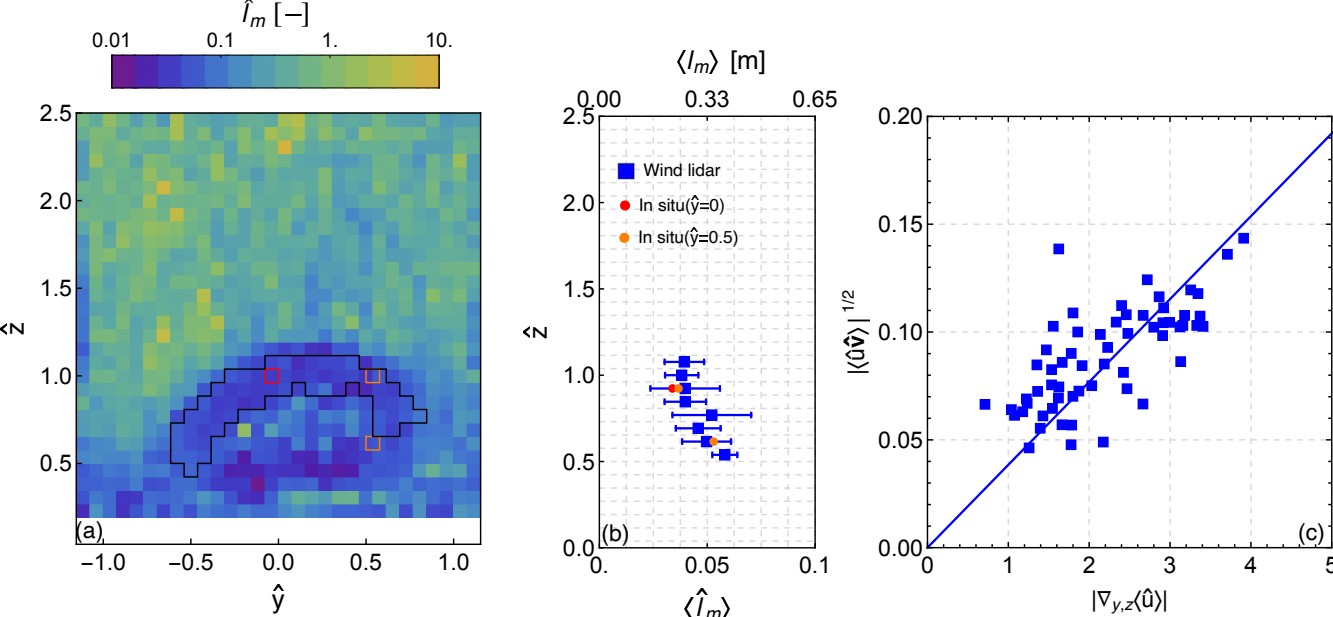

**Figure 6. Turbulence mixing length**. (a), Normalized mixing-length values ($\hat{l}_m$) along the scanning plane in the lee-side of the oak tree presented in a logarithmic scale. The solid black line marks the wake region where high gradients are observed. The position of the co-located *in situ* sensors (sonic anemometer) in that region is highlighted with red ($\hat{y}=0$), and orange ($\hat{y}=0.5$). (b), Vertical profile of the horizontally averaged mixing-length of the high gradient region of the wake (blue) and estimated mixing length values based on the in *in situ* (in red and orange). (c), Scatter plot between the vectors of the mean gradient and of the momentum flux of the transverse wind. The solid blue line depicts a linear least squared fit forced through zero.

which we use here to estimate a local length scale per grid cell. The results are presented in Fig.4a and highlight the strong reduction of the mixing length values between the free and wake flow.

We find that the horizontally averaged values of the normalized mixing length vary between 0.038 - 0.058, which is consistent with the estimation of the length scale using the sonic anemometer point measurements (presented in red and orange in Fig.6b). The estimation of the mixing length using the sonic anemometer measurements was also based on equation 5, by combining the momentum flux measurements of the sonic anemometer with the co-located measurement of the gradient of the mean wind speed measurements from the scanning wind lidars. In those grid cells with high $\langle \hat{u}'\hat{u}' \rangle$ variance, the covariance vector

estimations using the wind lidars have a maximum relative difference of -6% in comparison with the observations from sonic anemometers.

The mixing length is observed to be near constant in the top part of the crown ($\hat{z} > 0.8$), with an increasing trend with decreasing height. In the lower heights we use measurements only from one side of the crown, therefore the observed trend could be biased, since the crown is not homogeneously dense. Overall, we observe that the mixing length can be approximated

by a constant value in this near-wake region based on the high correlation between the terms of Eq. 5 (Pearson correlation

coefficient $r = 0.7$, Fig.6c). The average normalized mixing length is estimated to be equal to 0.038, which corresponds to 0.25 m.

This value is one order of magnitude lower than the mixing lengths estimated in dense (Seginer et al., 1976; Poggi et al., 2004) and sparse (Pietri et al., 2009) artificial and simplified canopies. However, our result is in close agreement with the mixing length values estimated in a wind tunnel study in the wake of a three-dimensional, fractal, tree-shaped structure (Bai et al., 2012). Yet, in that study a clear height dependence of the mixing length was found, which was attributed to the vertical variations of the fractal complexity of the tree. Here, we do not observe a similarly strong trend (Fig.6b). In the case of a mature, open-grown tree, leaves and branches of varying scales are in-homogeneously distributed in the crown volume. However, around the edges of the tree, the crown is characterized by similar density and size of the branches and twigs. This physical characteristic could explain the strong correlation between the length of the covariance and gradient vectors that supports the efficient description of the turbulence eddies in the wake edges using the mixing-length hypothesis. Hence, even if the fractal tree in Bai et al. (2012) was made as a simplification of a real tree, the effect of real trees on turbulence transport can be represented in a simpler way.

## 5    Discussion and Conclusion

In this study we have presented spatially distributed measurements of the wind vector in the near-wake region of a solitary, mature oak tree. The measurements were acquired using a scanning remote sensing system that consisted of three synchronized continuous-wave wind lidars. The synchronous scanning, the high sampling frequency and the short probe volumes, relative to the wind flow characteristics, enabled the measurements of the first and second order moments of the flow which depict the momentum fluxes between the free and wake flow.

From the three hour long period examined in this study only 101 scanning pattern iterations were finally selected in the analysis due to the wind direction requirement that was chosen. The period of sampling in each grid cell (26 seconds) is at least fives times larger than the time integral scale of the momentum fluxes in the wake of the tree (see Fig. C1 in Appendix C). This, in combination with the rapid sampling frequency of the wind lidars used here, enables the theoretical estimation of the systematic and random error in the calculated momentum fluxes based on Eq. 56 (systematic error) and Eq. 59 (random error) in Lenschow et al. (1994). We find that theoretically the largest contribution to be expected in due to random errors which are estimated to be approximately equal to 15%. The relative difference between momentum fluxes estimated using the wind lidar and sonic anemometer measurements in the wake area, in those locations where high variance is observed, was smaller than 20%, and this can explain the good agreement between the corresponding length scale estimations which are presented in Figure 6. A larger data sample would help to reduce the random error variance on both the estimations of the second order moments and of the corresponding momentum fluxes.

Using the selected dataset, we show that in the layer of high turbulence that separates the wake from the free flow the momentum fluxes of the streamwise component can be adequately predicted by the transverse gradient of the mean flow . This is a good news, since it means that relatively simple flow models that rely on the eddy-viscosity principle can be used to realistically

reproduce the effect of trees, similar to the one studied here, for applications in agriculture, erosion-prevention, air-pollution,
pollen-dispersion, as well as modelling of atmospheric transport at different scales. Realistic predictions of the momentum
fluxes using the Boussinesq hypothesis, are probably achievable also in the case of other small-scale tree configurations, as
well as sparse canopies, where the flow can be characterized by a super-position of multiple wakes behind each single tree.

The results presented in this study highlight the value of remote sensing systems based on three synchronously scanning
wind lidars to probe complex flows, since a very high spatial resolution of the heterogeneous flow field could be achieved.
Conventional *in situ* sensors such as sonic anemometers, would not be able to provide such spatial detail without distorting
the flow. The new capability to directly measure the three-dimensional turbulent fluxes over large planes, as demonstrated
here, can also be used to explore and describe other atmospheric flows that are difficult to correctly scale in wind tunnels.
Hence, the presented measurement technology enables new possibilities to explore and improve our description of the complex
earth-atmosphere interaction.

*Data availability.* The data used in this study are available upon request.

## Appendix A: Wind Conditions

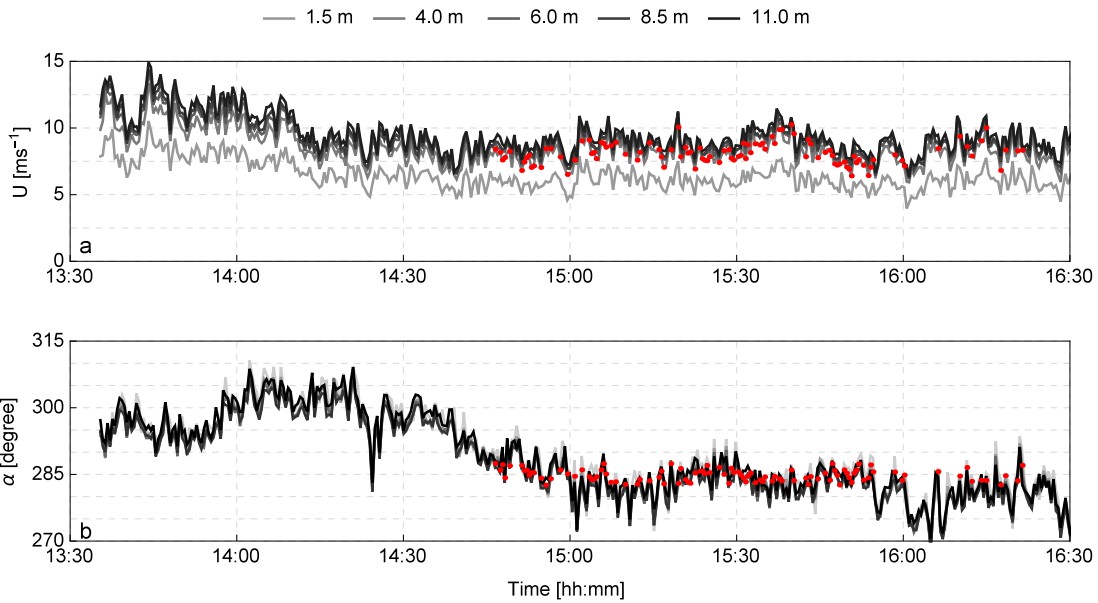

**Figure A1.** Time series of the 26-second mean wind speed (a) and direction (b) over a period of three hours between 13:30 – 16.30 (UTC+2) on 25-10-2017. The measurements were acquired by five sonic anemometers located on the $M_1$ mast at 1.5, 4.0, 6.0, 8.5 and 11.0 m above ground level.

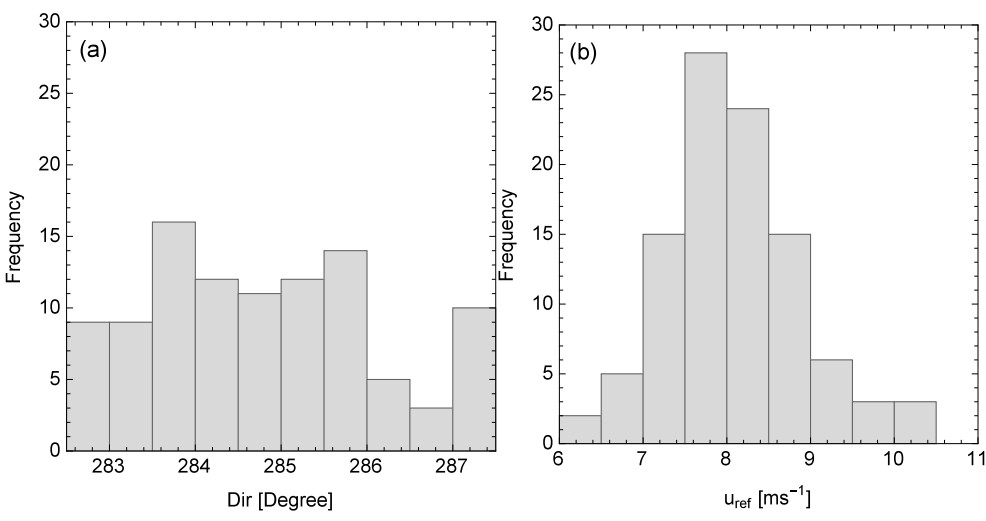

**Figure A2.** Histograms of the 26-second mean wind speed (a) and direction (b) over those periods when the mean wind direction was within the $282.5° - 287.5°$ sector. The measurements were acquired by the sonic anemometers located on the $M_1$ mast at 4.0 above ground level.

## Appendix B: Grid cell selection for the validation of the Boussinesq hypothesis

The validity of the Boussinesq's hypothesis was investigated in those grid cells characterized by high variance of the streamwise component $\langle \hat{u}'\hat{u}' \rangle$. The selection of those grid cells was performed by first averaging three vertical profiles of $\langle \hat{u}'\hat{u}' \rangle$ located in each of the two edges of the scanning plane in order to reconstruct the vertical variations of $\langle \hat{u}'\hat{u}' \rangle$ of the free flow (Fig.B1). Subsequently, all the measurements of the vertical plane were normalized by the reference $\langle \hat{u}'\hat{u}' \rangle$ profile. Finally, those grid cells with normalized $\langle \hat{u}'\hat{u}' \rangle$ values higher than the statistical outer fence of the normalized $\langle \hat{u}'\hat{u}' \rangle$ distribution were chosen for the analysis.

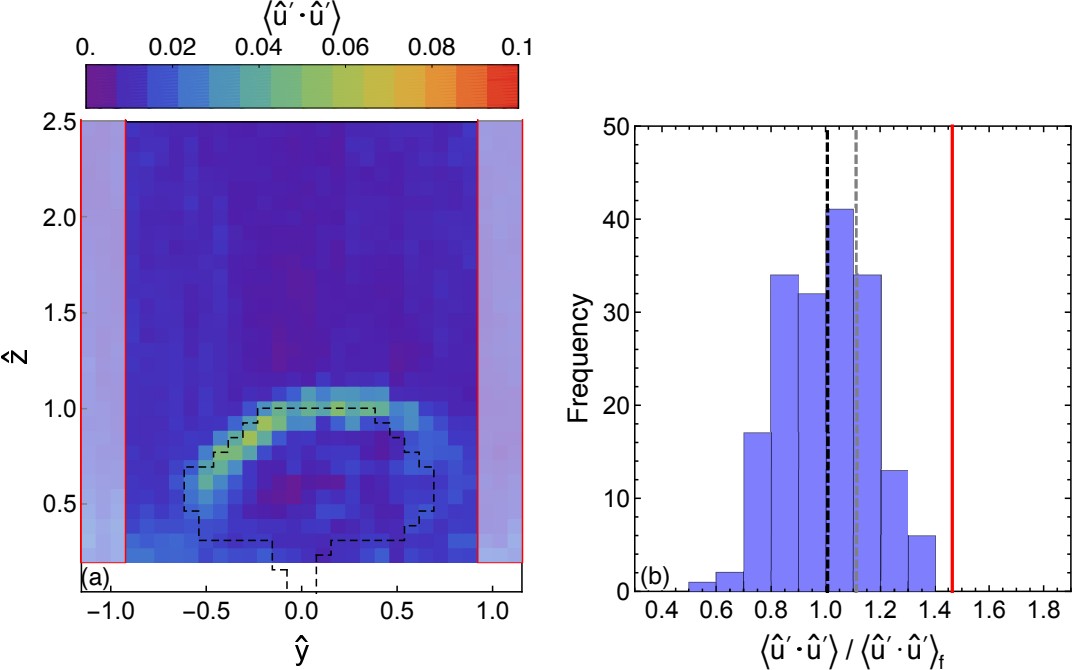

**Figure B1.** (a) Area used for the estimation of the reference variance of the free wind marked by the two red rectangles and (b) histogram of normalized variance $\langle \hat{u}'\hat{u}' \rangle/\langle \hat{u}'\hat{u}' \rangle_f$ in the selected grid cells and the corresponding the median (black dashed) and upper quantile (grey dashed), as well as the upper outer fence (red) values. The $\langle \hat{u}'\hat{u}' \rangle_f$ denotes the height dependent variance of the $u$ component of the free wind.

## Appendix C: Time integral scale of the momentum fluxes in the wake of the tree

Figure C1 presents the estimated time integral scale of the vertical and horizontal momentum fluxes. For the calculation there were used the time series of the wind measurements from the sonic anemometers in the $M_2$ meteorological mast. The time integral scale values in each height were estimated by integrating the autocorrelation function until the first time lag for which the autocorrelation function dropped to zero. We find that the time integral scale of the momentum fluxes in the wake of the tree (heights 4 and 6 m) are between $2.22 - 4.53$ seconds for the $\langle u'v' \rangle$ and between $0.17 - 0.32$ seconds for the $\langle u'w' \rangle$.

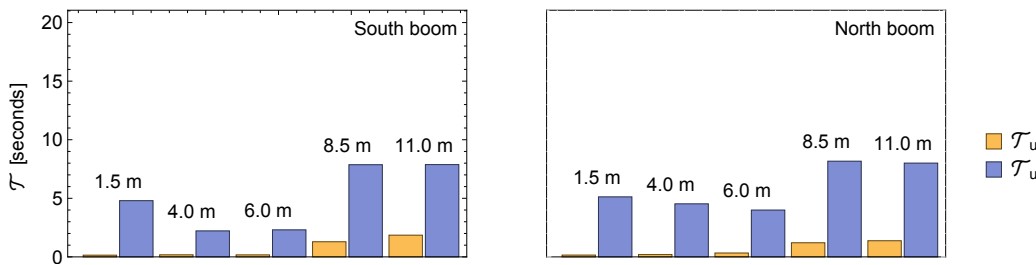

**Figure C1.** Time integral scale $\mathcal{T}$ of $\langle u'v' \rangle$ and $\langle u'w' \rangle$ estimated using the time series of the measurements acquired in each of the 10 sonic anemometers of the $M_2$ meteorological mast, found at the south (left) and north (right) side of the mast.

*Author contributions.* E.D. and J.M. conceived the experiment. N.A. planned and conducted the experiment, post-processed and analysed the acquired data and drafted the manuscript. E.D. and J.M. provided critical input and contributed to the interpretation of the results. All authors contributed to the writing of the final version of the manuscript.

*Competing interests.* The authors declare that they have no conflict of interest.

*Acknowledgements.* The Independent Research Fund Denmark is acknowledged for supporting this work via the *The Single Tree Experiment* project [Grant No. 6111-00121B]. Claus Brian Munk Pedersen and Per Hansen, research technicians in the Wind Energy Department of DTU, are acknowledged for their valuable support during the realization of the experimental campaign.

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
