# Peer review of "Wind lidars reveal turbulence transport mechanism in the wake of a tree"

_Atmospheric Chemistry and Physics, 2021_

## Referee Comment (RC1)

**Wind lidars reveal turbulence transport mechanism in the wake of a tree**

**Summary comment**

In this manuscript, a laser-based remote sensing instruments is used to analyze flow statistics in the wake of an open-grown oak tree. The subject is indeed interesting and relevant, given that the current knowledge on turbulence in the wake of trees is limited and that it profoundly influences exchange processes between the land surface and the atmosphere. The *Introduction* is appropriate, reads well, and is well referenced. The *Methodology* section is OK but could be improved (MC1 and MC2). The *Results* section starts off with the validation of Lidar measurements against the sonics, which is convincing and interesting but lacks important information (MC3-MC5). The authors then propose an interesting overview on spatially-distributed flow statistics, and conclude with an analysis of the eddy viscosity and mixing length concepts, which unfortunately comes across as flawed (MC6). The written english is good overall. I invite the authors to address my comments below before the manuscript can be considered for publication.

**Mayor comments**

1. L113. Was this a period of relatively statistically-stationary flow at the tree location? I ask because non-stationarity (e.g. strong accelerations and decelerations of the wind field) at a time scale comparable to the ones induced by the tree wake could lead to departures from the usual atmospheric turbulence. It would be useful to analyze the flow variability at scales comparable to the height of the tree over the local friction velocity too ensure that the considered periods are indeed stationary flow periods.

2. L131. Can you provide more details as to what the "corresponding distribution" refers to? Is this from 26' sampling at different locations? The processing procedure should be better described to enable an assessment of its impact on results as well as to enable others to reproduce these results in the future.

3. L145. This comparison is really interesting. I encourage the authors to provide a quantitative comparison (percentage values) for the second order moments as well. Since I can imagine that variations can be as high as 200% at certain locations, perhaps one can show the values and mention that these are within the observed uncertainty? This, again, would be very valuable information in my opinion, especially when considering the scope of the study. What's the impact of these "errors" (assuming the sonics are correct) on the eddy viscosity and mixing length quantities?

4. L174. Assuming that the along wind gradients are much smaller than the transverse wind gradients is a rather strong assumption in this "complex geometry" flow. Can the authors support this assumption anyhow?

5. Eq2. I recommend using standard index or vector notation as this expression is a bit confusing. Plus it seems to me that the eddy viscosity cannot be a scalar in this case but should rather be a first order tensor, otherwise this expression implies that $\overline{u'v'}/(du/dy) = \overline{u'w'}/(du/dz)$ (if I understood the expression correctly). I invite the authors to clarify this point.

6. L188. I am not sure what point the authors are trying to make here. The eddy viscosity is mathematically defined as a ratio of fluxes and mean wind gradients, and as such, it can indeed be used to describe the overall momentum flux within a plant canopy. Whether it makes physical sense though, that is another point. For example, in the presence of counter gradient fluxes, its value would be negative, which is unphysical and would lead to e.g. a blow up of simulations. Similarly, if the main flux is from large scale coherent structures, then the concept of eddy viscosity is not the right one, even if its value is positive. With their analysis, the authors have just shown that $K$ can be mapped to fluxes, but this is just a result of their mathematical definition. Further, it is not clear to me what percentage of the total momentum flux is really caused by the considered Reynolds stresses - can the authors quantify it? I bet that dispersive flux contributions might be significantly larger, i.e. $(\overline{uw} \gg \overline{u'w'})$, cause this flow is not statistically homogeneous and there is strong subsidence and flow three-dimensionality in the wake region. This also justifies why the authors have found a rather small mixing length in their studies. By the way, the authors can probably compute a good estimate of the total drag that the tree is exerting on the flow directly from the velocity map in Fig. 4(a). This would help determine the overall contributions of $\overline{u'w'}$ to the total drag.

**Minor comments**

1. L12. Perhaps better to say "extracting"?
2. L13. Cause → Can cause
3. L16. The increase in turbulence is not only because of increased wind gradients, but also via wake generation and via the adverse pressure gradient that they generate.
4. L38. Critical extension → Since the authors are not modifying the measurement instrument/methodology, perhaps it is better to say "a new application"?

---

## Author Response (AR1)

**ACP-2021-598 - Authors reply to the comments of the reviewers**

Nikolas Angelou, Ebba Dellwik and Jakob Mann

October 2021

**General answer by the authors**

We would like to thank both reviewers for their comments and suggestions to correct and improve this manuscript. We have tried to answer thoroughly to all their comments. Please find below the comments of both reviewers in black fonts and our corresponding answers in blue fonts.

**Reviewer 1**

**Major comments**

1. L113. Was this a period of relatively statistically-stationary flow at the tree location? I ask because nonstationarity (e.g. strong accelerations and deccelerations of the wind field) at a time scale comparable to the ones induced by the tree wake could lead to departures from the usual atmospheric turbulence. It would be useful to analyze the flow variability at scales comparable to the height of the tree over the local friction velocity too ensure that the considered periods are indeed stationary flow periods. The experiment lasted between 14:30 and 16:20 (Figure A1 in the original manuscript). The local mean friction velocity during that period, estimated using the sonic measurements at the height of 4 m at the upwind meteorological mast was approximately equal to 0.4 m/s during this measurement campaign, see Angelou et al. (2021). The tree height over the friction velocity corresponds to a time scale  $\tau_H$ , which is equal to 6.5 m / (0.4 m/s) = 16.25 seconds. In Figure 1, we show the time series of the amplitude of the wind vector at 4 m, using mean values over  $\tau_H$ . There was no significant wind speed trend over the 1 hour and 50 minutes of the experiment. Using a linear fit, we find a slope of less than 0.001 m/s per  $\tau_H$ , and, therefore, we assume that the investigated time series can be treated as stationary.

Figure 1: Time series (black) of the mean wind speed at 4 m, at a time scale  $\tau_H$  equal to the tree height over the friction velocity. The red line depicts the result of the linear fit.

The following sentence will be added after the line 114 of the original version: In order to assess the stationarity of the free flow, we define the time scale  $\tau_H = u_\star/H$ , where  $u_\star = (\langle u'w' \rangle^2 + \langle v'w' \rangle^2)^{1/4}$  was calculated from the 4 m sonic anemometer at the  $M_1$  mast and H is the tree height. Using the mean of the friction velocity  $(0.4 \text{ ms}^{-1})$ ,  $\tau_H = 16.25 \text{ s.}$  Over this time scale, a linear fit of the wind speed showed a slope of less than  $0.001 \text{ ms}^{-1}$  per  $\tau_H$ , and the time series of the free flow is, therefore, considered to be stationary.

2. L131. Can you provide more details as to what the "corresponding distribution" refers to? Is this from 26' sampling at different locations? The processing procedure should be better described to enable an assessment of its impact on results as well as to enable others to reproduce these results in the future.

The "corresponding distribution" refers to the distribution of the vertical component w of the wind vector in each grid cell. We initially tried to apply a filter on all the velocity components, but the only significant effect was found when applying it to w, which is due to geometry of the wind scanner setup. Each wind lidar measures the projection of the wind vector to the direction of the laser beam ("line-of-sight"). For the lower part of the measurement plane, the elevation angle of the laser beam direction gets close to 0 degree. This situation would correspond to trying to measure the vertical velocity component with a sonic anemometer, which only had near-horizontal transducer pairs. Any slight misalignment of the transducer pairs or noise in the line-of-sight wind speed measurements would result in a large error in the estimation of the vertical component.

The sentence of the original manuscript:

First, a filtering was applied in each 130 grid cell by treating as outliers those wind vector measurements with a vertical component w outside the inner and outer lower fence of the corresponding distribution.

Is now re-written to:

First, a filter was applied on the wind lidar data. The filtering was based on the calculation of the inner and outer fence of the distribution of w in each grid cell. Those wind vector estimations with a vertical component outside the two fences in each grid cell were treated as outliers and were not included in the analysis.

3. L145. This comparison is really interesting. I encourage the authors to provide a quantitative comparison (percentage values) for the second order moments as well. Since I can imagine that variations can be as high as 200% at certain locations, perhaps one can show the values and mention that these are within the observed uncertainty? This, again, would be very valuable information in my opinion, especially when considering the scope of the study. What's the impact of these "errors" (assuming the sonics are correct) on the eddy viscosity and mixing length quantities?

We agree that it is important to provide an assessment on the accuracy of the estimated  $2^{nd}$  order moments and also agree that this was not presented thoroughly in the original version of the article. The estimated values of the 2nd order moments, using the wind lidars and the sonic anemometers is presented in the Figure 3. The relative error in the longitudinal variance varies between -6% and -58%. The maximum relative error is found in the sonic anemometer at the center of the wake (sonic anemometer at 4 m, south boom at  $M_2$ ), where the turbulence is very low. The systematic underestimation of the longitudinal variance by the lidars can be explained by their larger probe volumes compared to the path lengths of the sonic anemometers. Hence, a part of the high-frequency variance is filtered out in the lidar data. The  $\langle u'w'\rangle^2$  and  $\langle v'w'\rangle$  fluxes are less affected by the lidar probe volumes, since the high-frequency co-spectrum drops faster than the power spectrum in the inertial subrange. For the momentum fluxes, we still find relative error values that are up to 128%, excluding the  $\langle u'w' \rangle$  observations at the lowest heights (1.5 m), which due to the geometry of the experimental setup (despite the filtering described above) are found to be very sensitive to random noise. Furthermore, we have not included the vertical momentum flux measurement at 4 m at the north boom of the M2 mast, where  $\langle u'w' \rangle$  was found to be equal to 0 m2s-2, leading to an unspecified relative error. These errors are a lot smaller in the case when the momentum fluxes have high values as it is the case in the wake of the tree. In those locations, which are the most important for our study, we find that all errors are smaller than 20%. For the fluxes, the disagreement between the sonic anemometers and the lidars, can be explained by the random error caused by the disjunct sampling of the velocity field (see answer 2 to reviewer 2, below). A longer time series would reduce the random error of all the second order moments. Concerning how the disagreement between the two measurements affect the further results, we refer to Figure 6, where we see a good agreement between the estimated length scales by the sonic anemometers and the lidars. For this calculation, it is the error on the length of the stress vector that matters  $\{\langle u'v' \rangle, (\langle u'w' \rangle\}$ . For the high-gradient region, this error is typically significantly less than 20 % which explains the good agreement between the estimated length scale values using the wind lidar and sonic anemometer measurements. The following sentences have been added in:

Section 4.1: The relative error in the longitudinal variance varies between -6% and -58%. The maximum relative error is found in the sonic anemometer at the center of the wake (sonic anemometer at 4 m, south boom at  $M_2$ ), where the turbulence is very low. As far as it concerns the momentum fluxes, we find overall relative error values that are up to 128% above the tree, but the error in the wake is limited to 20%. This does not include the vertical momentum fluxes at the lowest heights (1.5 m), which due to the geometry of the experimental setup, are found to be very sensitive to random noise. Furthermore, we have not included the vertical momentum flux measurement at 4 m at the north boom of the  $M_2$  mast, since the correlation between the longitudinal and vertical component was found to be equal to 0 m2s-2, leading to an unspecified relative error.

Discussion: The relative difference between momentum fluxes estimated using the wind lidar and sonic anemometer measurements in the wake area, where high variance is observed, was smaller than 20%, and this can explain the good agreement between the corresponding length scale estimations which are presented in Figure 6. A larger data sample would help to reduce the random error variance on both the estimations of the second order moments and of the corresponding momentum fluxes.

4. L174. Assuming that the along wind gradients are much smaller than the transverse wind gradients is a rather strong assumption in this "complex geometry" flow. Can the authors support this assumption anyhow?

Dellwik et al. (2019) studied the wake generated by the same tree in our study using numerical simulations, that were extensively and successfully validated by sonic anemometer observations. According to that study, (please see Figure 10 in (Dellwik et al., 2019), the along-wind gradients are much smaller than the gradients across the wake at the downwind distance of  $1.3 \times H$ . In addition, we have estimated the along wind gradients ( $\frac{\partial U}{\partial x}, \frac{\partial V}{\partial x}$  and  $\frac{\partial W}{\partial x}$ ) in the vicinity of the 10 sonic anemometers on the M2 mast, since the scanning plane by the wind lidars was located approximately 1.3 m upwind of the from the sonic anemometers. The values that we find are one order of magnitude lower than the values of transverse gradient. Therefore, we think that the results of our study are not biased by disregarding the along wind gradient. We elaborate more on this topic in our answer to the 7th comment of Reviewer 2.

5. Eq2. I recommend using standard index or vector notation as this expression is a bit confusing. Plus it seems to me that the eddy viscosity cannot be a scalar in this case but should rather be a first order tensor, otherwise this expression implies that u0v0/(du/dy) = u0w0/(du/dz) (if I understood the expression correctly). I invite the authors to clarify this point.

After the recommendation of both reviewers we have changed the formulation of Equation 2. Please see our reply to the Comment 7 of the second reviewer for a detailed answer, along with the corresponding changes that we did in the revised version of the document.

6. L188. I am not sure what point the authors are trying to make here. The eddy viscosity is mathematically defined as a ratio of fluxes and mean wind gradients, and as such, it can indeed be used to describe the overall momentum flux within a plant canopy. Whether it makes physical sense though, that is another point. For example, in the presence of counter gradient fluxes, its value would be negative, which is unphysical and would lead to e.g. a blow up of simulations. Similarly, if the main flux is from large scale coherent structures, then the concept of eddy viscosity is not the right one, even if its value is positive. With their analysis, the authors have just shown that K can be mapped to fluxes, but this is just a result of their mathematical definition. Further, it is not clear to me what percentage of the total momentum flux is really caused by the considered Reynolds stresses - can the authors quantify it? I bet that dispersive flux contributions might be significantly larger, i.e.  $(uw \cdot u_0w_0)$ , cause this flow is not statistically homogeneous and there is strong subsidence and flow three-dimensionality in the wake region. This also justifies why the authors have found a rather small mixing length in their studies. By the way, the authors can probably compute a good estimate of the total drag that the tree is exerting on the flow directly from the velocity map in Fig. 4(a). This would help determine the overall contributions of u0w0 to the total drag.

The point of the presented analysis is to show that it makes sense to use the eddy-viscosity formulation to predict fluxes from the gradients in the wake of the tree. The results show that in the region around the periphery of the wake, in which momentum transfer processes between the free and the wake flow take place, the vector of the fluxes is anti-parallel to the mean transverse gradient. We study the flux-gradient relation in many individual sub-areas that each have a size of  $0.5 \text{ m} \times 0.5 \text{ m}$ . These distributed observations are treated as point observations and the spatial variability within each area is assumed to be negligible. This assumption is justified by the good agreement with the sonic anemometer observations in the wake of the tree. Dispersive fluxes occur where a spatial average is taken over an area with strong spatial variability. Indeed, if the wake of the tree could only be represented by a single grid point in a numerical model, the dispersive component of the flux would be highly significant. Concerning the last comment about the total drag of the tree, we refer another article of ours (Angelou et al., 2021), which was published recently. In that study, we estimated the drag force on the tree from the momentum deficit in the wake. We also quantified that the contribution of the momentum fluxes is less than 10% of the overall drag induced by the tree to the flow.

**Minor comments**

- 1. L12. Perhaps better to say "extracting"? Corrected
- 2. L13. Cause  $\rightarrow$  Can cause Corrected
- 3. L16. The increase in turbulence is not only because of increased wind gradients, but also via wake generation and via the adverse pressure gradient that they generate. We have changed the word "leads" with the one "contributes", therefore the revised sentence is: This, in turn, contributes to generation of turbulence ..
- 4. L38. Critical extension → Since the authors are not modifying the measurement instrument/methodology, perhaps it is better to say "a new application"? Corrected

**1 Reviewer 2**

**General Comment**

In this manuscript the authors perform turbulence measurements in the field, in the wake of an isolated large tree. This is a quite complete work, done with relatively heavy measurement devices: a multi-lidar and several sonic anemometers placed on two meteorological masts. This results in an important database which is statistically analyzed in this manuscript. This is an important work, the observation efforts needed to record this data base is very appreciable. It will be a useful database for the community. I have not seen a data availability statement: it would be useful to provide this information. Below I have several suggestions and comments.

Answer: Thank you for the comments and the suggestions. The data are going to be accessible upon request. A statement will be added in the revised manuscript.

**Specific comments**

1. I did not understand figures 2a-b, concerning the orange dots. I understand that there are two meteorological masts, one of them (M2) being on the red dot. Why is the orange dot in a different location? The  $M_2$  mast was instrumented with in total 10 sonic anemometers, two at each height. The orange dots correspond to the sonic anemometer locations on the Northerly boom on the  $M_2$  mast, whereas the red dots corresponds to the locations on the Southern boom. We re-write the lines 103–105 of the first version of the paper as:

The sonic anemometers were installed on booms pointing towards the direction of  $200^{\circ}$  relative to North on both masts. To get a higher coverage of the complex wind field in the tree wake, the  $M_2$ mast was instrumented with five additional sonic anemometers, on opposing booms pointing towards  $20^{\circ}$  relative to North. In order to clarify this point, we have also re-formulated the text in the caption of Figure 2:

Drawings of the top (left) and the front (right) views of the experimental setup used in this study, where the locations of the tree, the short-range WindScanners (WS1, WS2 and WS3), the scanning pattern (blue) and the locations of the sonic anemometers on the up- (M1) and down-wind (M2) meteorological masts, are depicted. The locations of the 10 sonic anemometers on the opposing booms on the M2 mast are indicated with red and orange color, respectively.

2. For each grid, it is indicated that there is approximately 25 iterations per 10 minutes period. Since the whole data set is recorded during a 3 hours periods, I understand that there are 18x25 = 450 values at each grid point, to perform the statistics. Is this correct? If correct it does not seem to be a large sample size to perform statistics. This point should be clarified in the manuscript and discussed in the discussion section.

We agree with the reviewer that ideally more measurements would be required to estimate the wind statistics with a higher accuracy. From the 3 hours period examined in this study only 101 scanning pattern iterations are selected finally in the analysis due to the wind direction requirement that was chosen. However, despite the disjunct sampling performed here, we argue that the resulting statistics are still sufficiently accurate.

In order to assess theoretically the accuracy of the calculated  $2^{nd}$  order moment we refer to the work of Lenschow et al. (1994), where it is shown that for the estimation of  $2^{nd}$  order moments and fluxes, it is important to use a sensor that can acquire observations instantly and not averaging over time. Then the systematic and random errors in the estimated fluxes is going to be dependent on the number of measurements (N) and the time between consecutive observations  $(\Delta)$ . When  $\Delta$  is a lot larger than the time integral scale of the measured parameter then the systematic relative to the ideal flux (denoted

here as  $F_i$ ) can be estimated using equation 56 in (Lenschow et al., 1994):

$$\frac{F_i - \langle u'_1 u'_i \rangle_s}{F_i} = \frac{\Delta}{T} \quad \text{with} \quad i = 2,3 \tag{1}$$

and equation 59 in (Lenschow et al., 1994) for the random error:

$$\sigma_F^2 = \mu_f \frac{\Delta}{T}.$$
 (2)

In the two equations above, T is the length of the period examined,  $\mu_f$  is the variance of the flux time series (see Lenschow et al. (1994) equation (45) and equations thereafter) when T is approximating infinity and  $\langle u_1 u'_i \rangle_s$  denotes the mean of the measured momentum flux over the period T and with a sample separation  $\Delta$ .

Using the time series of the sonic anemometer in the M2 meteorological mast, we estimated the time integral scale of the momentum fluxes, by integrating the autocorrelation function until the first time lag for which the autocorrelation function dropped to zero. The results are presented in Figure 2. In this figure we can see that the time integral scales of the momentum fluxes in the wake of the tree (heights 4 and 6 m) are between 2.22 - 4.53 seconds for the  $\langle u'v' \rangle$  and between 0.17 - 0.32 seconds for the  $\langle u'w' \rangle$ .

Figure 2: Time integral scale of  $\langle u'v' \rangle$ ,  $\langle u'w' \rangle$  estimated using the time series of the measurements acquired in each of the 10 sonic anemometers of the M2 meteorological mast.

Since, the wind lidars are measuring very fast (approximately 200 Hz) and the time of the disjunct sampling is lot larger than the time integral scales, then we can use the Equations 1 and 2 to derive an estimation on the theoretical systematic and random error in the measured second order moments. We find that despite the limited amount of data the systematic error of the measured momentum fluxes is about 1% (26 seconds over 2626 (26x101) seconds). The random error is between 6% and 15% for the case of the vertical and transverse momentum flux, respectively.

- 3. I don't understand the normalisation indicated in line 131: "the wind speed measurements of each iteration were normalized by the corresponding 26-second mean wind speed". The normalisation should be the same for all data. If normalization depends on the samples, one needs to understand what are the statistics of these normalization values. This should be clarified. The data acquired during one iteration of the scanning pattern (26 seconds) were normalized by the mean wind speed (over 26 seconds), using the measurements of the sonic anemometer at the  $M_1$  mast. This way a time-dependent scaling was applied to the time series in order to have a constant mean value of the upwind conditions at 4 m, over a time scale equivalent to the scanning duration of the wind lidars.
- 4. The stresses  $\langle u'w' \rangle$  and  $\langle v'w' \rangle$  are estimated, as well as normal stresses. Since instantaneous values are taken every 25 seconds, it is not the real stress which is estimated (the real stress would need

to resolve turbulent scales, of the order of seconds or lower). The estimates  $\langle u'w' \rangle$  and  $\langle v'w' \rangle$  are done using coarse-gained estimates of u', v', w', as their fluctuations are likely to be much smaller than real values of stresses. This should be mentioned in the text and also discussed in the discussion section.

As we discuss in our answer to the  $2^{nd}$  comment of the reviewer the accurate estimation of the second order moments is not solely depending on the time of the disjunct time sampling. We agree that ideally we should have more measurements in order to reduce the random error in our estimated values. We will add the following text to *Discussion* section.

From the 3 hours period examined in this study only 101 scanning pattern iterations are selected finally in the analysis due to the wind direction requirement that was chosen. The period of sampling in each grid cell (26 seconds) is a lot larger than the time integral scale of the momentum fluxes in the wake of the tree. This enables the theoretical estimation of the systematic and random error in the calculation of the momentum fluxes based on the the mathematical formulation presented in Lenschow et al. (1994). We find that the largest contribution to be expected in due to random errors which are estimated to be approximately equal to 15%. The relative difference between momentum fluxes estimated using the wind lidar and sonic anemometer measurements in the wake area, where high variance is observed, was smaller than 20%, and this can explain the good agreement between the corresponding length scale estimations which are presented in Figure 6. A larger data sample would help to reduce the random error variance on both the estimations of the second order moments and of the corresponding momentum fluxes.

- 5. The gradients along y and z of the mean velocity are estimated (fig 4b,c). How this gradient is estimated numerically should be indicated in the text (central difference, some smoothing is done with a kernel?). The gradients are estimated numerically by:
  - (a) first estimating numerically the forward difference between neighbouring grid cells along the y and z axis using:

$$\frac{\partial \langle \hat{u} \rangle}{\partial \hat{y}} = \frac{\langle \hat{u} \rangle [i+i] \rangle - \langle \hat{u} \rangle [i]}{\hat{y}[i+1] - \hat{y}[i]} \text{ and } \frac{\partial \hat{u}}{\partial \hat{z}} = \frac{\langle \hat{u} \rangle [i+i] - \langle \hat{u} \rangle [i]}{\hat{z}[i+1] - \hat{z}[i]}$$
(3)

(b) and subsequently by applying a linear interpolation and finding the corresponding value of the gradient in the y, z coordinates of the grid cell

The following sentence has been added in the manuscript in the Section 4.2: The gradients were estimated by first calculating the forward difference of the mean longitudinal wind speed between neighbouring grid cells in the y and z direction, and subsequently by finding the corresponding value in the coordinates of each cell using linear interpolation.

6. Figure 4c plots one normal stress. It could be interesting to plot the 2D kinetic energy and also to estimate the turbulence intensity, by dividing fluctuations by the mean velocity, to obtain a percentage.

During the data analysis performed in this study we estimated both the turbulence intensity (TI) and the turbulence kinetic energy (TKE), but we decided not to include them in the manuscript. Figure 3 presents both plots, where it can be seen the following: a. there is a clear increase in the TI along the wake and this increase is spatially inhomogeneous, which is in accordance with the inhomogeneous porosity of the crown of the tree. b. The TKE, which here is equal to 1/2( + < v'v' > + < w'w' >), is higher along the periphery of the crown of the tree. At the lower measuring heights ( $\hat{z}

---

## Referee Report (RR1)

**Wind lidars reveal turbulence transport mechanism in the wake of a tree**

**Summary comment**

In this revised submission the authors have successfully addressed the majority of my previous concerns. The *Methodology* section has substantially improved: The experimental setup is well described and the data processing procedure is clear and reproducible. The *Results* section has also improved, but the manuscript would benefit from a more usable definition of the area of interest in the wake of the tree (mc1) and from a more rigorous mathematical treatment and notation (mc2 and several other mcs). Overall this is an interesting study that makes use of a truly unique experimental setup and data; the modeling community will be able to readily take advantage of its data and findings.

**Minor comments**

- 1. "We investigate this further by selecting grid points with high u'u' comparing to the undisturbed flow" -> Why are the author's limiting their analysis to this region of the flow? The way the selection of this region of interest appears as quite convoluted and I would imagine that the modeling community would benefit much more from this analysis if the validity of the eddy viscosity assumption was assessed in the whole wake region, rather than in a thin (albeit dynamically important) layer. This is mostly a recommendation to improve the quality of the manuscript.
- 2. Related to the comment above. I find that several of the assumptions that the authors have put forth (i.e., streamwise gradients of vertical and cross-stream velocities are negligible) are not necessary to support their claim of validity of the Boussinesq hypothesis. Why, for example, not taking the full velocity gradient tensor and momentum flux tensor into consideration and verify the relative alignment of their eigenvectors? It would also be useful if the authors could report the magnitude of the velocity gradients in the streamwise direction when compared to the streamwise ones. This, again, is mostly a recommendation.
- 3. L201. strain -> strain rate (and elsewhere)
- 4. L215. Equation (3) is saying nothing about the stream-wise gradients of the vertical and cross-stream velocities, so this remark is not correct. In other words, if stream-wise gradients of the vertical and cross-stream velocities were not equal to zero, equation 3 would still be valid since it describes a relation between different quantities. Please rephrase.
- 5. Caption of Fig. 5. I recommend using index notation for the wind gradient vector as well, since otherwise it is difficult to relate gradients and corresponding momentum fluxes.
- 6. L229. The simplest parameterization for the eddy viscosity is assuming it is a constant. The mixing length is a level up in terms of complexity. Perhaps it is better to say "a relatively simple..."?
- 7. EQ4. Are the authors referring to the L2 norm of the velocity gradient tensor? In this case I would recommend using this symbol:  $\|\cdot\|_2$  to avoid confusion.

8. EQ5. Since the authors are only considering i = 1, 2, I recommend writing out the full expression for hte momentum flux, i.e.  $\sqrt{(u'v')^2 + (u'w')^2}$ . Same elsewhere.

---

## Author Response (AR2)

**ACP-2021-598 - Authors reply to the comments of the $2^{nd}$ review**

Nikolas Angelou, Ebba Dellwik and Jakob Mann

January 2022

We thank both reviewers for their constructive comments and suggestions and for their very interesting scientific regarding the work presented in this manuscript. Their comments were helpful both for finding weak points that needed improvement and for improving the communication of the results. We have thanked both reviewers in the *Acknowledgement* section.

Please find below our answers (in blue fonts) to the comments of the reviewers (in black fonts).

**1 Reviewer 1**

**Summary comment**

In this revised submission the authors have successfully addressed the majority of my previous concerns. The Methodology section has substantially improved: The experimental setup is well described and the data processing procedure is clear and reproducible. The Results section has also improved, but the manuscript would benefit from a more usable definition of the area of interest in the wake of the tree (mc1) and from a more rigorous mathematical treatment and notation (mc2 and several other mcs). Overall this is an interesting study that makes use of a truly unique experimental setup and data; the modeling community will be able to readily take advantage of its data and findings.

**1.1** Minor Comments**

"We investigate this further by selecting grid points with high u0u0 comparing to the undisturbed flow"
-> Why are the author's limiting their analysis to this region of the flow? The way the selection of this
region of interest appears as quite convoluted and I would imagine that the modeling community would
benefit much more from this analysis if the validity of the eddy viscosity assumption was assessed in
the whole wake region, rather than in a thin (albeit dynamically important) layer. This is mostly a
recommendation to improve the quality of the manuscript.

We agree with the reviewer that from a flow modelling perspective ideally the assessment of the validity of the eddy viscosity hypothesis should take place alone the whole cross-section of the wake, but our data is not of sufficiently high quality to do this. The main limitation is the measurement error relative to the magnitude of both the mean wind gradient and the momentum fluxes in locations with a low-turbulence flow. This limitation originates both from the characteristics of the experimental setup (i.e. wind lidar probe lengths) and the random errors in the estimated values of the wind vector that are attributed to the limited length of the acquired data set. The following sentence has been added in the Discussion section after the line 284 of the revised version, to clarify this point:

The reduction of random errors, in combination with smaller probe lengths of the wind lidar, would enable the study of the relation between the momentum fluxes and the mean gradients even in the center of the wake, where very small gradients are observed. Furthermore, we changed the following sentence:

Line 221: We investigate this further by selecting grid points with high  $\langle u'u' \rangle$  comparing to the undisturbed flow (for more information regarding the grid selection we refer to the Appendix B). We chose these grid cells both because they represent the area where the mixing takes places.

**as:**

Line 221: We investigate this further by selecting grid points with high  $\langle \hat{u}' \hat{u}' \rangle$  comparing to the undisturbed flow (for more information regarding the grid selection we refer to the Appendix B). We chose these grid cells because they represent the area where the mixing of momentum between the free and wake flow takes place.

2. L201. strain -> strain rate (and elsewhere).

**Corrected**

3. L215. Equation (3) is saying nothing about the stream-wise gradients of the vertical and cross-stream velocities, so this remark is not correct. In other words, if stream-wise gradients of the vertical and cross-stream velocities were not equal to zero, equation 3 would still be valid since it describes a relation between different quantities. Please rephrase.

We do not agree with the reviewer in this point. In this study we focus on the transport mechanism of the longitudinal momentum. For this purpose, as we state in the text we construct a momentum vector from the following two components:

$$\langle u_1' u_i' \rangle = \nu_T \frac{\partial u_1}{\partial x_i}, \text{ where } i = 2, 3.$$
 (1)

However, this equation originates from the Reynolds stress tensor, according to which:

$$\langle u_i' u_j' \rangle - \frac{1}{3} \langle u_k' u_k' \rangle \delta_{ij} = -\nu_T \left( \frac{\partial U_i}{\partial x_j} + \frac{\partial U_j}{\partial x_i} \right).$$
(2)

According to Equation 2 the transverse and vertical momentum fluxes are equal to:

$$\langle u_1' u_2' \rangle = -\nu_T \left( \frac{\partial u_1}{\partial x_2} + \frac{\partial u_2}{\partial x_1} \right) \tag{3}$$

and

$$\langle u_1' u_3' \rangle = -\nu_T \left( \frac{\partial u_1}{\partial x_3} + \frac{\partial u_3}{\partial x_1} \right).$$
 (4)

With our statement in line 215, we want to argue on where do we base our assumption that the contribution of the terms  $\frac{\partial u_2}{\partial x_1}$  and  $\frac{\partial u_3}{\partial x_1}$  can be considered negligible in the case of the flow examined in our study. We propose the following rewriting of the text in lines 215 – 219 to clarify this point:

With the above equation we want to express the relation between the momentum flux and mean gradient. This expression originates from Eq. 2, when the along wind gradients of the vertical  $\frac{\partial \langle u_2 \rangle}{\partial x_1}$  and transverse components  $\frac{\partial \langle u_3 \rangle}{\partial x_1}$  are considered to be negligible. We base this assumption on the estimated values of the along wind gradients  $\left(\frac{\partial \langle u_2 \rangle}{\partial x_1} \text{ and } \frac{\partial \langle u_3 \rangle}{\partial x_1}\right)$  based on the wind lidar and sonic anemometer measurements at the 10 locations, where sonic anemometers were found on the  $M_2$  mast.

4. Caption of Fig. 5. I recommend using index notation for the wind gradient vector as well, since otherwise it is difficult to relate gradients and corresponding momentum fluxes.

We agree that the current caption can be confusing. The sentence Direction of (a) the mean gradient  $\left| \left( \frac{\partial \langle u \rangle}{\partial y}, \frac{\partial \langle u \rangle}{\partial z} \right) \right|$  and (b) the covariance  $\langle u'_1 u'_i \rangle$  vectors. has been rewritten as: Direction of (a) the mean gradient  $\frac{\partial \langle \hat{u}_1 \rangle}{\partial \hat{x}_i}$  and (b) the covariance  $\langle \hat{u}'_1 \hat{u}'_i \rangle$  vectors.

5. L229. The simplest parameterization for the eddy viscosity is assuming it is a constant. The mixing length is a level up in terms of complexity. Perhaps it is better to say "a relatively simple. . . "?

Corrected: The simplest parameterization of the eddy viscosity  $\nu_T$ ... is now re-written as: A relatively simple parameterization of the eddy viscosity  $\nu_T$ ...

6. EQ4. Are the authors referring to the L2 norm of the velocity gradient tensor? In this case I would recommend using this symbol:  $\|\cdot\|_2$  to avoid confusion.

The norm in the velocity gradient tensor in Equation 4 refers to the Euclidian norm. However, we would prefer to keep the notation already used. We suggest the following addition in line 233: where  $\left| \left( \frac{\partial \langle u \rangle}{\partial y}, \frac{\partial \langle u \rangle}{\partial z} \right) \right|$  represents the Euclidean norm of the transverse gradient of the mean longitudinal wind.

7. 8. EQ5. Since the authors are only considering i = 1, 2, I recommend writing out the full expression for the momentum flux, i.e.  $\sqrt{(u'v')^2 + (u'w')^2}$  Same elsewhere.

We agree with this suggestion. The equation:

$$l_m = \frac{\sqrt{\left|\langle u_1' u_i' \rangle\right|}}{\left|\left(\frac{\partial \langle u \rangle}{\partial y}, \frac{\partial \langle u \rangle}{\partial z}\right)\right|}$$

is rewritten as:

$$l_m = \frac{(\langle u'v'\rangle^2 + \langle u'w'\rangle^2)^{1/4}}{\left| \left( \frac{\partial \langle u \rangle}{\partial y}, \frac{\partial \langle u \rangle}{\partial z} \right) \right|}.$$

**2 Reviewer 2**

I am fine with most of the revisions which have been done following my review. I have only some remaining comments on the Eddy-viscosity check which was done in this work. The new equation 3 is a 2D vector alignment, which is derived from the tensorial relation 2. The Boussinesq eddy-viscosity relation is the tensorial one. We have (3) => (2) but the reverse is obviously false. Two minor changes should be done in the manuscript.

1. In the abstract, about the "validity of the eddy-viscosity hypothesis", replace by "the validity of a 2D vectorial relation derived from the eddy-viscosity hypothesis".

We agree with this suggestion. We changed the sentence accordingly.

2. In the text, Line 237: replace "supports the validity of the eddy-viscosity hypothesis" by "supports a two-dimensional vectorial alignment between vectors, derived from the tensorial eddy-viscosity hypothesis".

We agree with this suggestion. The sentence:

Using the same criterion as Schmitt (2007), we find that the observed relative direction of the two vectors supports the validity of the eddy-viscosity hypothesis.

is now re-written as:

Using the same criterion as Schmitt (2007), we find that the observed relative direction of the two vectors supports a two-dimensional vectorial alignment, derived from the tensorial eddy-viscosity hypothesis.

**References**

Schmitt, F. G.: About Boussinesq's turbulent viscosity hypothesis: historical remarks and a direct evaluation of its validity, Comptes Rendus Mécanique, 335, 617 – 627, https://doi.org/https://doi.org/10.1016/j. crme.2007.08.004, 2007.